# Structural, Spectroscopic, and Biological Characterization of Novel Rubidium(I) and Europium(III) Co-Doped Nano-Hydroxyapatite Materials and Their Potential Use in Regenerative Medicine

**DOI:** 10.3390/nano12244475

**Published:** 2022-12-17

**Authors:** Nicole Nowak, Dominika Czekanowska, John M. Reeks, Rafal J. Wiglusz

**Affiliations:** 1Institute of Low Temperature and Structure Research, Polish Academy of Sciences, Okolna 2, 50-422 Wroclaw, Poland; 2Department of Animal Biostructure and Physiology, Wroclaw University of Environmental and Life Sciences, Norwida 25, 50-375 Wroclaw, Poland

**Keywords:** nanosized hydroxyapatite, Eu^3+^ ion-doping, Rb^+^ ion-doping, regenerative medicine, tissue engineering, photoluminescence, in vitro cell-culture study

## Abstract

This research investigates hydrothermally synthesized hydroxyapatite nanoparticles doped with rubidium(I) and europium(III) ions. Investigation focused on establishing the influence of co-doped Eu^3+^ and Rb^+^ ions on hydroxyapatite lattice. Therefore, structural, and morphological properties were characterized via using X-ray powder diffraction (XRPD), infrared spectroscopy (FT-IR), and scanning electron microscopy (SEM), as well as transmission electron microscopy (TEM) techniques. Furthermore, this investigation evaluates the impact of various Rb^+^ ion doping concentrations on the distinct red emission of co-doped Eu^3+^ ions. Hence, luminescence properties of the obtained materials were evaluated by measuring emission excitation, emission spectra, and luminescence decays. As established by numerous studies, synthetic hydroxyapatite has excellent application in biomedical field, as it is fully biocompatible. Its biocompatible makes it highly useful in the biomedical field as a bone fracture filler or hydroxyapatite coated dental implant. By the incorporation of Eu^3+^ ions and Rb^+^ ions we established the impact these co-doped ions have on the biocompatibility of hydroxyapatite powders. Therefore, biocompatibility toward a ram’s red blood cells was evaluated to exclude potential cytotoxic features of the synthesized compounds. Additionally, experimental in vitro bioactive properties of hydroxyapatite nanoparticles doped with Rb^+^ and Eu^3+^ ions were established using a mouse osteoblast model. These properties are discussed in detail as they contribute to a novel method in regenerative medicine.

## 1. Introduction

Developments in chemistry and nanotechnology have allowed for the production of synthetic hydroxyapatites. These materials exhibit the necessary mechanical properties, biocompatibility, and bioactivity to form a strong bond with living bone tissue. Due to their osteoinductive properties, hydroxyapatite ceramics are widely used as scaffolding, as filling material in polymeric biocomposite implants, and as coatings for metallic implants [1,2,3]. Through photo-stabilization of rare earth ions’ luminescence, these materials can also be used as biomedical sensors [4]. The use of specific, biologically active dopants can improve the biocompatibility of hydroxyapatite-based compounds and enhance regenerative properties towards bone, cartilage, and even nervous tissue [5].

Rare-earth lanthanide ions (e.g., europium(III) ions) are known to have remarkable luminescence properties and are widely used in spectroscopic applications. Due to having similar ionic radii and chemical properties, lanthanides as well as rare-earth and alkaline elements can be easily substituted for calcium (Ca^2+^) in hydroxyapatite crystalline structure [6,7]. Europium(III) ions substituting for calcium(II) ions in the hydroxyapatite lattice introduces the possibility of obtaining new materials with better photoluminescence properties [8,9]. Furthermore, hydroxyapatites doped with europium(III) ions have excellent biocompatibility and biodegradability. They, therefore, can be used in biomedical applications as fluorescent labelling materials [10,11]. 

Research shows rubidium(I) ions are advantageous in biological applications. Neuroprotective properties of rubidium ions (RbCl) have been reported against pentylenetetrazole (PTZ) induced seizures. This is suggested to be mediated via NMDA (N-methyl-D-aspartate) receptors and NO (nitric oxide) pathway in hippocampus [12]. Another independent study confirmed Rb^+^ ions (RbCl) exhibit anti-depressant activity by modulating NO pathway in mice [13]. Studies also indicate that biologically safe Rb^+^ ions (RbCl) impaired osteoclastogenesis ovariectomized (OVX) and titanium (Ti) particle-induced calvaria osteolysis mouse model and instead stimulated osteogenesis in vitro [14]. Therefore, a biologically active dopant such Rb^+^ ions can improve biocompatibility of hydroxyapatite-based compounds and enhance osteoinductive properties towards bone cells. 

The aim of this research was to obtain hydroxyapatite (HAp) based compounds co-doped with Eu^3+^ and Rb^+^ ions and investigate their nanostructure as well as their improved luminescence and biological properties. For the first time, hydroxyapatite-based composites co-doped with Eu^3+^ and Rb^+^ ions were obtained in two series of powder compounds with the chemical formula Ca_10 − (0.1 + x)_Eu_0_._1_Rb_x_(PO_4_)_6_(OH)_2_ and Ca_10 − (0.2 + x)_Eu_0.2_Rb_x_(PO_4_)_6_(OH)_2_ (where x = 0.5; 1; 2; 3; 5; 7 mol%). This study confirmed that the obtained compounds exhibit enhanced photo-luminescence properties and increased biocompatibility toward the mouse osteoblasts cell line. 

## 2. Materials and Methods

### 2.1. Synthesis Method

Two series of novel hydroxyapatite based nanopowder materials doped with Eu^3+^ and Rb^+^ ions with the chemical formula Ca_10 − (0_._1 + x)_Eu_0.1_Rb_x_(PO_4_)_6_(OH)_2_ and Ca_10 − (0.2 + x)_Eu_0.2_Rb_x_(PO_4_)_6_(OH)_2_ (where x = 0.5; 1; 2; 3; 5; 7 mol%) were synthesized via hydrothermal method. The substrates used in synthesis were respectively Ca(NO_3_)_2_∙4H_2_O (99.0–103.0% Alfa Aesar, Karlsruhe, Germany), Eu_2_O_3_ (99.99% Alfa Aesar, Karlsruhe, Germany), (NH_4_)_2_HPO_4_ (>99.0% Acros Organics Chemicals-Thermo Fisher Scientific, Waltham, MA, USA) and RbCl (≥99.0% Sigma Aldrich, Saint Louis, MO, USA). Stoichiometric amounts of Ca(NO_3_)_2_∙4H_2_O, (NH_4_)_2_HPO_4_, and RbCl were dissolved, separately, in deionized water. Next, the stoichiometric amount of Eu_2_O_3_ was digested in HNO_3_ (≥65.0%, Sigma-Aldrich, Saint Louis, MO, USA) to obtain water soluble europium nitrate (Eu(NO_3_)_3_). This was then recrystallized three times (100 °C in deionized water) to eliminate the remaining of HNO_3_. The obtained Eu(NO_3_)_3_ was then dissolved in deionized water and mixed with aqueous Ca(NO_3_)_2_∙4H_2_O and RbCl solutions. The substrates were introduced to the aqueous (NH_4_)_2_HPO_4_ solution. For example, the synthesis reaction for 1 g of Ca_9.8_Eu_0.1_Rb_0.1_(PO_4_)_6_(OH)_2_, where Ca:P ratio equals 1.66 involves dissolving 2.2681 g of Ca(NO_3_)_2_∙4H_2_O in 25 mL of deionized water, 0.0119 g of RbCl in 5 mL of deionized water and 0.7765 g of (NH_4_)_2_HPO_4_ in 10 mL of deionized water. Preparation of Eu(NO_3_)_3_ begins with dissolving 0.0172 g of Eu_2_O_3_ in 5 mL of deionized water and 200µL of HNO_3_ (≥65.0%, Sigma-Aldrich, Saint Louis, MO, USA) to obtain water soluble Eu(NO_3_)_3_. After recrystallization, Eu(NO_3_)_3_ was dissolved in 5 mL of deionized water. When all reagents were prepared water solutions of Ca(NO_3_)_2_∙4H_2_O, RbCl and Eu(NO_3_)_3_ were mixed in the Teflon vessel and then water solution of (NH_4_)_2_HPO_4_ was added. The pH was then adjusted to pH = 9.0 using ammonia (NH_3_∙H_2_O 25% Avantor, Gliwice, Poland). The reaction was carried out in a microwave reactor (ERTEC MV 02-02, Wrocław, Poland) for 90 min at a temperature of 245–250 °C under autogenous pressure (45–50 bar). After the synthesis, materials were dried for 1 day at 90 °C. Finally, powders were heat treated at 500 °C for 3 h, the temperature was set to increase at the rate of 3 °C per minute.

### 2.2. Structural Characterization 

X-ray diffraction (XRD) was used to characterize the crystal structure of the synthesized hydroxyapatite powders co-doped with Eu^3+^ and Rb^+^ ions. XRD patterns were performed using a PANalytical X’Pert Pro X-ray diffractometer (Malvern, United Kingdom) with Ni-filtered Cu Kα radiation (U = 40 kV, I = 30 mA) in the 2θ range of 5–70°. The step for XRD analysis was as 0.0263 and the time per step was estimated 4.36 s per step. The XRD-recorded patterns were compared with the reference hydroxyapatite pattern from the Inorganic Crystal Structure Database (ICSD). The chemical components of the synthesized materials were identified by the inductively coupled plasma optical emission spectrometer (ICP-OES) Agilent 720 instrument (Santa Clara, California, United States). The samples were prepared for ICP by dissolving 100 mg of nanopowder material in 2 mL of HNO_3_ (≥65.0%, Sigma-Aldrich, Saint Louis, MO, USA) at the temperature 120 °C and by gradual adding of deionized water to the final volume of 50 mL. The content of P(V), Eu(III), Rb(I) elements were measured in the solutions diluted 20 times and the concentration of Ca(II) was measured in the solution diluted 500 times. For these measurements, the ICP OES method (Agilent, model 720) was used to analyze 3 prepared samples of the solution in parallel. These results were compared with standard curves in the concentration range of 0.05 to 5.00 mg/mL for Ca, Eu, Rb elements and 100 to 200 mg/mL for phosphorus. To evaluate the presence of phosphate groups in the structure of obtained compounds, IR spectra were measured in the range of 4000–400 cm^−1^ (mid-IR) at 295 K. Attenuated total reflectance (ATR-FT-IR) measurements were recorded at a resolution of 4 cm^−1^ (32 scans) using a Nicolet iS50 infrared spectrometer (Thermo Fisher Scientific, Waltham, MA, USA). The elemental mapping analysis of the selected sample was determined by using an FEI Nova NanoSEM 230 scanning electron microscope (FEI, Hillsboro, Oregon, United States) operating at an acceleration voltage in the range 3.0–15.0 kV and spot size of 4.0–4.5. The morphology and nanostructure of obtained composites were determined using FEI Nova NanoSEM 230 scanning electron microscope (FEI, Hillsboro, Oregon, United States) operating at an acceleration voltage in the range 3.0–15.0 kV and spot size 4.0–4.5. The samples were evenly coated with a layer of graphite prior to these observations. Illustration of the map was automatically generated by the microscope software. The morphology, size, and structure of nanopowder materials were visualized by via HRTEM (High Resolution Transmission Electron Microscopy), using a Philips CM-20 Super Twin microscope (FEI, Hillsboro, Oregon, United States), operated at 200 kV. The selected hydroxyapatite based powders were prepared via dispersion in methanol. Then, a drop of suspension was deposited on a copper microscope grid covered with perforated carbon and each sample was imaged.

### 2.3. Luminescence Properties

Excitation and emission spectra as well as luminescence kinetics, of novel Ca_10 − (0.1 + x)_Eu_0.1_Rb_x_(PO_4_)_6_(OH)_2_ and Ca_10 − (0.2 + x)_Eu_0.2_Rb_x_(PO_4_)_6_(OH)_2_ (where x = 0.05; 0.1; 0.2; 0.3; 0.5; 0.7) nanopowder materials were collected via FLS980 fluorescence spectrometer (Edinburgh Instruments, Livingston, United Kingdom). A 450 W Xenon Lamp was used as the excitation light source during the excitation and emission spectra records. Specific excitation wavelengths were isolated using a 300 mm monochromator equipped with a holographic grating (1800 grooves per mm, blaze of 250 nm) in conjunction with the lamp. Measuring the luminescence kinetics involved utilizing a microsecond flashlamp (uF2) excitation source with a Hamamatsu R928P photomultiplier tube detector (Shizuoka, Japan). The excitation spectra and luminescence kinetics were excited by 393.5 nm light. Luminescence kinetics were observed at 616.7 nm emission which corresponds to the most intense electric dipole transition of Eu^3+^ ions (from level *^5^D_0_ → ^7^F_2_* level) [15,16,17].

### 2.4. Biological Properties

#### 2.4.1. Preparation of Sample Suspensions

Before the preparation of the colloidal solutions, obtained powder materials were sterilized under UV light for 30 min. Then, stocks of nanosized hydroxyapatites doped with Eu^3+^ and Rb^+^ ions were prepared by forming suspensions of compounds in distilled water in the concentration 1 mg per mL. Then, each stock was bath-sonicated for 1 h at RT. Freshly prepared colloids were sterilized under the UV light for 30 min before being used in biological experiments.

#### 2.4.2. Cell Culture and Viability Assay

Mouse osteoblasts (7F2) cell line was maintained in high glucose Alpha Modified Eagle Medium (α-MEM) with L-glutamine (Biowest, Nuaillé, France) and supplemented with 200 U/mL penicillin and 200 µg/mL streptomycin and 10% heat-inactivated fetal bovine serum (FBS, South America origin, Biowest, Nuaillé, France). The cell line was incubated in standard conditions in a humid atmosphere of 95% air and 5% CO_2_ at 37 °C. The cell line was passaged three times before the experiments were performed. 

To evaluate biocompatibility of the obtained compounds MTT viability assay was performed on mouse osteoblasts cell line (7F2, ATCC, Manassas, Virginia, United States). 7F2 cells were seeded at a density of 1 × 104 cells per well in a 96-well plate. Cells were then treated with two series of compounds at three different concentrations: 10 µg/mL, 50 µg/mL, and 100 µg/mL. Cultures of treated and untreated cells were incubated for 24 h in the standard conditions. After that time, the medium containing the tested compounds was removed and cells were washed out once with sterile PBS. Then, freshly prepared MTT (Sigma-Aldrich, Saint Louis, MO, USA) reagent (0.5 mg/mL) was dissolved in sterile PBS and added to the cells. The negative control group was established as untreated cells and was equal 100% of viable cells. Cells were incubated in a humidified atmosphere of 95% air and 5% CO_2_ for 3 h at 37 °C. After that time, MTT solution was removed, and formazan crystals were dissolved by adding isopropanol. Absorbance was read at 560 nm and 670 nm (background reference). This experiment was conducted three times. The viability of used cell lines was estimated using the following formula:$$Cell\;viability\;=\frac{sample\;absorbance}{control\;absorbance}\times 100$$

To establish biocompatibility of rubidium(I) ions and its potential positive effect on proliferation rate of mouse osteoblasts cell line (7F2), RbCl (rubidium chloride) solution was prepared. 10 mM solution of RbCl was obtained by dissolving RbCl powder (≥99.0% Sigma Aldrich, Saint Louis, MO, USA) in sterile PBS and then filtrated via sterile syringe filter (pore size 0.2 µm). The 10 mM of RbCl was further sterilized under UV light for 30 min. This prepared solution was used in the viability assay (MTT). Cells were seeded in the density 1 × 104 per well in 96-well plate, then RbCl was added to cells to obtain different final concentration of the solution: 1.0; 1.25; 2.5; 3.75 and 5.0 mM. The negative control group was established as untreated cells having 100% viability of cells. After 24 h of incubation the MTT assay as well as the calculations were performed as it was described above.

#### 2.4.3. Evaluation of Mouse Osteoblast Morphology

To evaluate cell morphology mouse osteoblasts (7F2) were seeded in the density 5 × 104 per well at the 24-well plate and incubated with both series of compounds in the highest tested concentration 100 µg/mL. After 24 h incubation, cells were washed with sterile PBS, then fresh PBS was added to each well and Invitrogen™ ReadyProbes™ Cell Viability Imaging Kit (Blue—live/Green—dead) (Invitrogen-Thermo Fisher Scientific, Waltham, MA, USA) was used to visualize dead–live cell ratio. Cell morphology was observed using an Invitrogen™ EVOS™ FL Digital Inverted Fluorescence Microscope (×10 magnification) (Thermo Fisher Scientific, Waltham, MA, USA).

#### 2.4.4. Hemolysis Assay

Sterile and defibrinated sheep blood (Pro Animali, Wrocław, Poland) was washed out 3 times in sterile PBS and ultimately suspended in sterile PBS (Biowest, pH 7.4) at a ratio of 1:1. Ca_10 − (0.1 + x)_Eu_0.1_Rb_x_(PO_4_)_6_(OH)_2_, and b) Ca_10 − (0.2 + x)_Eu_0.2_Rb_x_(PO_4_)_6_(OH)_2_ powders, where x equals 0.5; 1; 2; 3; 5; 7 mol% of Rb^+^ ions were tested towards sheep red blood cells at concentrations of 50 µg/mL and 100 µg/mL. To establish a positive control, sheep erythrocytes were combined with 10% SDS (sodium dodecyl sulfate) (Sigma Aldrich, Saint Louis, MO, USA) and treated as 100% of hemolysis, negative control was obtained by mixing sheep erythrocytes with sterile PBS. After 24 h of incubation at 37 °C, positive and negative control as well as red blood cell samples treated with selected hydroxyapatite-based composites were centrifuged (5000 RPM, 5 min) to obtain supernatant. The optical density was then measured at 540 nm (Varioscan Lux, Thermo Fisher Scientific, Waltham, MA, USA). The hemolysis percentage was calculated using the formula below:$$Hemolysis=\frac{sample\;absorbance-negative\;control\;absorbance}{positive\;control\;absorbance-negative\;control\;absorbance}\times 100$$

Red blood cell morphology and the integrity of cell membranes were observed via an Invitrogen™ EVOS™ FL Digital Inverted Fluorescence Microscope (×10 magnification). Sheep erythrocytes were prepared as described above and treated with selected compounds (only with the highest concentration 100 µg/mL); positive and negative control was also prepared. After 24 h of incubation at 37 °C, red blood cells were centrifuged (5000 RPM, 5 min). The supernatant was gently removed, and cell precipitate was suspended with sterile PBS at a ratio of 1:1. A blood smear was prepared by transferring 5 µL of sample onto a microscope slide and using a coverslip to obtain the smear. 

## 3. Results and Discussion

### 3.1. Characterization of Structure and Morphology

Two series of hydroxyapatite-based compounds with chemical formula Ca_10 − (0.1 + x)_Eu_0.1_Rb_x_(PO_4_)_6_(OH)_2_ and Ca_10 − (0.2 + x)_Eu_0.2_Rb_x_(PO_4_)_6_(OH)_2_, where x equals 0.5, 1, 2, 3, 5, 7 mol% of Rb^+^ were obtained via hydrothermal method and their structure and morphology were accurately analyzed (Figure 1A,B). X-ray diffractograms of obtained nanopowders were evaluated and compared to the standard ICSD database diffractogram pattern of hydroxyapatite crystalline material ICSD 262004. The results for investigated compounds clearly showed strict correspondence to the hydroxyapatite database pattern ICSD 262004. X-ray diffractograms of hydroxyapatite co-doped with Eu^3+^ and Rb^+^ ions were also juxtaposed with pure hydroxyapatite doped with Eu^3+^ ions (1 and 2 mol%). This comparison sought to establish the influence of Eu^3+^ dopants influence the hydroxyapatite crystal’s structure. Experimental XRD patterns of first and second series of hydroxyapatites co doped with 1 mol% of Eu^3+^ ions and x mol% of Rb^+^ ions and with 2 mol% of Eu^3+^ ions and x mol% of Rb^+^ ions (x = 0.5, 1, 2, 3, 5, 7) correspond to the pure hydroxyapatite structure. For all nanopowder series of materials containing 1 mol% of Eu^3+^/x mol% of Rb^+^ ions and 2 mol% of Eu^3+^/x mol% of Rb^+^ ions signals, especially in the range from 32° to 34° matches with distinctive phosphate groups of crystalline hydroxyapatite structure ICSD 262004 (Figure 1A,B). Interestingly, in the range of 29.51° a secondary phase can be observed in both doped series of nanopowders (Figure 1A,B), which can correspond to the signal of RbH_2_PO_4_ (ICSD 20312) [18,19]. This signal occurs in both series when rubidium(I) molar concentration is at 2 mol% or greater. Signal from the secondary phase increases with increasing rubidium ion concentration (Figure 1A,B). One possible explanation is that the rubidium precursor (RbCl) appears in higher concentration during the synthesis process. Therefore, with the presence of (NH_4_)_2_HPO_4_, a secondary RbH_2_PO_4_ phase appears. The specific conditions (temperature, pressure) of the reaction may also influence the appearance of this secondary phase [20]. The appearance of the second phase may also result from ionic radius mismatches in the hydroxyapatite structure: calcium(II) ions Ca^2+^ (C.N. = 9) = 1.18 Å and Ca^2+^ (C.N. = 7) = 1.06 Å, europium(III) ions Eu^3+^ (C.N. = 9) = 1.12 Å and Eu^3+^ (C.N. = 7) = 1.01 Å and rubidium(I) ions Rb^+^ (C.N. = 9) = 1.51 Å [21,22]. Since Rb^+^ ion has a larger ionic radius (0.148 nm) than Ca^2+^ ion (0.099 nm), the incorporation of Rb^+^ ion into the hydroxyapatite crystalline structure may lead to the lattice relaxation and eventually lead to changing unit cell parameters [23]. 

The FT-IR spectra of two series of obtained nanopowders Ca_10 − (0.1 + x)_Eu_0.1_Rb_x_(PO_4_)_6_(OH)_2_ and Ca_10 − (0.2 + x)_Eu_0.2_Rb_x_(PO_4_)_6_(OH)_2_, (where x equals 0.5, 1, 2, 3, 5, 7) confirmed crystalline hydroxyapatite structure through the presence of distinctive active vibrational bands, which correspond to phosphate groups (PO_4_^3−^) and hydroxyl groups (OH^−^) (Figure 2A,B). There is a characteristic double degenerate bending mode (ν_2_) of the P–O–P at 561.1838 cm^−1^ in the samples HAp 1 mol% Eu^3+^ and x mol% Rb^+^ ions and at 561.1838 cm^−1^ in the samples HAp with 2 mol% Eu^3+^ x mol% Rb^+^ ions. Triply degenerate bending mode (ν_4_) of the P–O bonds are visible at 599.7532 cm^−1^ for samples which contain HAp 1 mol% Eu^3+^ and x mol% Rb^+^ ions and at 599.2711 cm^−1^ for compounds with 2 mol% Eu^3+^ and x mol% Rb^+^ ions [24,25]. Moreover, the presence of non-degenerative symmetric stretching mode (ν_1_) of P–O bond at 1019.677 cm^−1^ and at 962.3051 cm^−1^ can be observed in HAp based compounds with 1 mol% Eu^3+^, x mol% of Rb^+^ ions and at 1021.605 cm^−1^ and at 963.2693 cm^−1^ for second series of HAp based compounds with 2 mol% Eu^3+^, x mol% of Rb^+^ ions. The triply degenerate asymmetric stretching mode (ν_3_) of the P–O bond are detected at 1088.137 cm^−1^ for hydroxyapatite nanomaterials with 1 %mol Eu^3+^, x mol% of Rb^+^ ions, but also at 1089.102 cm^−1^ for the second series of hydroxyapatite nanopowder materials co-doped with 2%mol Eu^3+^ x mol% of Rb^+^ ions [26,27]. Additionally, a narrow vibrational band can be seen at 3572.485 cm^−1^ for HAp co-doped with 1 %mol Eu^3+^, x mol% of Rb^+^ ions. A narrow band is also seen at 3572.968 cm^−1^ for HAp co-doped with 2% mol Eu^3+^, x mol% of Rb^+^ ions and it is seen due to stretching frequencies of hydroxyl groups OH^−^ of the surface-absorbed water [28].

The XRD results show evidence of a second phase (Figure 1A,B), especially among the samples with a concentration of co-doped rubidium(I) ions more than 2 mol%. By using Rietveld refinement and X-ray diffractometry results it is possible to evaluate the structural details of the samples, e.g., phase quantities, size and shape of crystallites, unit cell dimensions, atomic coordinates/bond lengths, and vacancies in the structure of the measured sample [29,30]. Through the use of FT-IR, it is possible to analyze many frequency components in the desire sample like for instance vibrational bands from phosphate groups (PO_4_^3−^) and hydroxyl groups (OH^−^) in hydroxyapatite structure [31]. Therefore, further investigation evaluated the molar concentrations of rubidium(I), europium(III), calcium(II) and phosphorous(V) via ICP-OES technique. This technique allows us to identify and quantify elements within matrix of the tested sample (Table 1) [32]. The ICP-OES measurements showed that actual content of desired elements is highly consistent with theoretical assumptions. The concentration of Eu^3+^ ions in the investigated samples is identical to the theoretical formulas. Additionally, it is clear (Table 1), that calcium(II) ions in the hydroxyapatite structure are nicely substituted by rubidium(I) and europium(III) ions among all measured samples. However, the concentration of rubidium(I) ions is slightly greater than theoretical values, as is the concentration of phosphorus. These results can along with X-ray diffractograms indicate the appearance of the second phase of RbH_2_PO_4_ (Figure 1A,B) [18,19]. Further confirmation of the presence of the desired elements in the materials was provided by SEM image and SEM-EDS mapping of the representative sample: Ca_9.1_Eu_0.2_Rb_0.7_(PO_4_)_6_(OH)_2_ (Figure 3). EDS maps confirm the presence of all theoretical elements like calcium(II), phosphorous(V), europium(III), rubidium(I) and oxygen(II). Moreover, all these elements are evenly distributed over the whole surface of hydroxyapatite based crystalline sample. Transmission Electron Microscopy technique was used to evaluate the nanostructure of the materials. In the Figure 4, TEM images clearly confirmed nanosized structure of two selected samples: Ca_9.8_Eu_0.1_Rb_0.1_(PO_4_)_6_(OH)_2_, Ca_9.7_Eu_0.1_Rb_0.2_(PO_4_)_6_(OH)_2_. In agreement with the Selected Area Electron Diffraction (SAED) technique, both sample materials showed excellently developed spotty rings, which highly correspond with crystalline structure of hydroxyapatite powder [33,34].

### 3.2. Investigation of Luminescence Properties

The results obtained by X-ray diffractometry (Figure 1A,B), clearly showed the crystalline hydroxyapatite structure appearance among the obtained nanopowders. However, there is an observed second phase for the hydroxyapatite-based compounds doped with 1 and 2 mol% of Eu^3+^ ions and co-doped with 3 mol% and more of Rb^+^ ions. As mentioned earlier, the presence of the second phase may indicate the appearance of RbH_2_PO_4_ in the mentioned samples. The XRD results seem to be confirmed also by Inductive Coupled Plasma Mass Spectroscopy (Table 1). Therefore, emission and excitation spectra for all obtained samples were recorded to establish whether the presence of the assumed RbH_2_PO_4_ has an influence on the luminescence properties of rubidium(I) and europium(III) co-doped nano-hydroxyapatite powder materials. 

The presence of europium(III) ions incorporated into the structure of hydroxyapatites was confirmed with luminescence studies. Samples were excited with 393.5 nm light to obtain emission spectra characteristic for Eu^3+^ ions. High quality emission spectra in the range of red-orange light were obtained for both 1% and 2% Eu^3+^ ions doped hydroxyapatites and co-doped with various concentration of Rb^+^ ions (Figure 5A,B). For the first series of compounds (Ca_10 − (0.1 + x)_Eu_0.1_Rb_x_(PO_4_)_6_(OH)_2_), five peaks are present as shown in Figure 5A. These peaks are present at 573.7 nm; 586.7 nm (Rb_x ≤ 3_)/588.8 nm (Rb_x ≥ 5_); 616.7 nm; 650.3 nm; 697.5 nm (Rb_x ≥ 5_)/700.2 nm (Rb_x ≤ 3_). The second series of obtained materials (Ca_10 − (0.2 + x)_Eu_0.2_Rb_x_(PO_4_)_6_(OH)_2_) show five emission peaks in Figure 5B. These are at wavelengths of 573.6 nm; 586.7 nm (Rb_x ≥ 3_)/ 589.6 nm (Rb_x ≤ 2_); 616.7 nm; 650.9 nm (Rb_x ≥ 3_)/654.2 nm (Rb_x ≤ 2_); 697.6 nm (Rb_x ≥ 3_)/700.5 nm (Rb_x ≤ 2_). These peaks characterize the transition from the excited level of ^5^D_0_ to the levels of ^7^F_0–4_ of Eu^3+^ ions [15]. The transitions are assigned respectively as: ^5^D_0_ → ^7^F_0_, ^5^D_0_ → ^7^F_1_, ^5^D_0_ → ^7^F_2_, ^5^D_0_ → ^7^F_3_, ^5^D_0_ → ^7^F_4_, with increasing wavelength value [15,35,36]. Figure 5A,B, show the most intense peak corresponding to the ^5^D_0_ → ^7^F_2_ transition. The peaks corresponding to this transition are observed at wavelengths in the range of 605–630 nm, with a maximum intensity at 616.7 nm [9,10,15,37]. Furthermore, the strictly forbidden transition of Eu^3+^ ion form the level ^5^D_0_ to ^7^F_0_ is well observed. The appearance of forbidden transition, according to the Judd-Ofelt theory, is a perfect example of the violation of selection rules. The most possible explanation of such Judd-Ofelt theory breakdown is that this transition is caused by crystal field perturbation, known also as J-mixing. Forbidden transition from level ^5^D_0_ to ^7^F_0_ can be also due to mixing of low-lying the charge-transfer states [38,39]. Intriguingly, some correlation exists between these two mechanisms, since J-mixing is increased by strong crystal-field effects, indicating these two effects can be interdependent [15,39]. The observation of the transition 0–0 indicates that Eu^3+^ ions occupy sites with the local symmetry of C_n_, C_nv_, C_s_, where “n” equals 1, 2, 3, 4, 6. This agrees with the theory that transition occurs only in the point symmetries that contain the linear terms in the crystal field Hamiltonian [38,39]. Nonetheless, three distinct splits of the transition ^5^D_0_ to ^7^F_0_ are observed among all obtained nanopowders, as shown in Figure 5A,B. Results indicate that Eu^3+^ ions can occupy three different crystallographic sties in the hydroxyapatite-based host lattice [40]. It is worth mentioning that the occupancy of the crystallographic sites changes when concentration of Rb^+^ ions increase. This is especially notable when rubidium ions concentration is equal to or greater than 2 mol%. Changes in the transition 0–0 observed for obtained nanopowder materials (Figure 5A,B) may indicate, that by substituting Ca^2+^ by Eu^3+^ and by Rb^+^ creates a charge imbalance, leading to lattice distortion [38]. The charge imbalance and difference between ionic radii of Ca^2+^, Eu^3+^ and Rb^+^ ions, as mentioned in paragraph 3.1., can lead to various defects creation in the hydroxyapatite host lattice. Cationic dopants such Eu^3+^ ions or Rb^+^ ions may typically substitute into two distinct Ca^2+^ sites in the hydroxyapatite structure: Ca_1_ and Ca_2_ with the site symmetry C_3_ and C_s_, respectively. Analyzing 0–0 transition can help determine which site Ca_1_ or Ca_2_ is preferable by Eu^3+^ ions [41]. Among both groups of obtained compounds, Eu^3+^ ions clearly showed more affinity to occupy the Ca_2_ site in samples which contain less than 5 mol% of Rb^+^ ions (Figure 5A,B). The results indicate that doped Rb^+^ ions clearly affect structure of the hydroxyapatite lattice and, therefore,Eu^3+^ occupancy sites in the host lattice. This leads to changes of the ^5^D_0_ → ^7^F_0_ transition. The appearance of the ^5^D_0_ to ^7^F_1_ directly refers to crystal-filed splitting of the ^7^F_1_ level [15,17]. Observation of magnetic dipole transition ^5^D_0_ → ^7^F_1_ showed the occurrence of more than three distinct lines indicative of the presence of more than one equivalent site for the europium(III) ions. All representative samples of Eu^3+^ and Rb^+^ ions co-doped hydroxyapatite nanopowder materials exhibit such features (Figure 5A,B). Nonetheless, according to the literature magnetic dipole transition is nearly independent of alteration in the local environment of Eu^3+^ ions in the crystal lattice. This work shows some changes of the transition are observed for the samples which contain 5 mol% and more of Rb^+^ ions. Hence, such reshaping of the ^5^D_0_ → ^7^F_1_ transition may indicate lattice disruption of theses representatives [36]. Hypertensive transition, known as the ^5^D_0_ → ^7^F_2_ transition, presents similar tendency with respect to the shape alterations, which can be observed in the Figure 5A,B. Contrary to magnetic dipole transition, hypertensive transition is strongly affected even by slight changes in the local environment of Eu^3+^ ions in the host lattice [15,36,42]. Therefore, analysis of the reshaped transition leads to the conclusion that Rb^+^ ions strongly affect the host lattice of obtained nanopowders. This is especially notable among the samples which are co-doped with 3 mol% and more of Rb^+^ ions.

The excitation spectra were measured in the wavelength range of 250–550 nm, for which the emission was monitored at 616.7 nm. The first series of compounds Ca_10 − (0.1 + x)_Eu_0.1_Rb_x_(PO_4_)_6_(OH)_2_, where x = 0.5; 1; 2; 3; 5; 7 mol% showed peaks from transitions distinct to Eu^3+^ ions. These are transitions ^7^F_0_ → ^5^H_(3–7)_, ^7^F_0_ → ^5^D_4_ and ^7^F_0_ → ^5^L_8_ as well as transitions ^7^F_0_ → ^5^G_(2,3)_ and ^7^F_0_ → ^5^L_7_. Moreover, for this group of nanopowders, ^7^F_0_ → ^5^L_6_, ^7^F_0_ → ^5^D_2,_ and ^7^F_0_ → ^5^D_1_ transitions were also recorded (Figure 6A). Interestingly, among the first series of compounds, samples co-doped with 1 mol% of Eu^3+^ ions and 3, 5, and 7 mol% of Rb^+^ ions exhibited additional transitions from the level ^7^F_0_ → ^5^F_(4–1)_ and ^7^F_0_ → ^3^P_0_. These transitions were recorded only for these three compounds of first series (Figure 6A). More notably, pure hydroxyapatite doped only with 1 mol% of Eu^3+^ ions lacked these transitions (Figure 6A). The intensities of respective transitions change and exhibit dependence on the rubidium(I) concentration. The results show enhanced intensities of individual transitions ^7^F_0_ → ^5^H_(7–3)_, ^7^F_0_ → ^5^D_4_, ^7^F_0_ → ^5^L_8_, ^7^F_0_ → ^5^G_(2,3)_, ^7^F_0_ → ^5^L_7_, ^7^F_0_ → ^5^L_6_, ^7^F_0_ → ^5^D_2_, and ^7^F_0_ → ^5^D_1_ as Rb^+^ ion concentration increases. This trend is especially notable for the following transitions: ^7^F_0_ → ^5^D_4_, ^7^F_0_ → ^5^L_8_, ^7^F_0_ → ^5^L_6_, ^7^F_0_ → ^5^D_2_, and ^7^F_0_ → ^5^D_1_. In the spectra recorded for samples co-doped with 1 mol% of Eu^3+^ and 0.5–7 mol% of Rb^+^ ions, an intense peak is visible at approximately 270 nm. This distinct transition corresponds to the charge transfer (CT) of an electron between the ionized oxygen atom and the europium ion (O_2_^−^→Eu^3+^) [17,36,42]. Our results are consistent with other results which provided data of charge transfer between O_2_^−^ and Eu^3+^ ions in the hydroxyapatite structure. As shown in Figure 6A, there is a trend of increasing CT from O_2_^−^ to Eu^3+^ as rubidium(I) ion concentration increases. The spectra obtained for second series of compounds showed similar peaks representative of transitions distinct to Eu^3+^ ions. These transitions are: ^7^F_0_ → ^5^H_(7–3)_, ^7^F_0_ → ^5^D_4_, and ^7^F_0_ → ^5^L_8_, ^7^F_0_ → ^5^G_(2,3)_, and ^7^F_0_ → ^5^L_7_ as well as transitions ^7^F_0_ → ^5^L_6_, ^7^F_0_ → ^5^D_2_, and ^7^F_0_ → ^5^D_1_ (Figure 6B). As opposed to the first series of compounds, the second series showed transition from the level ^7^F_0_ → ^5^F_(4–1)_ and ^7^F_0_ → ^3^P_0_ in all samples. This may indicate that increased concentration of Eu^3+^ enhances these transitions. The spectra also showed an increased intensity of CT from oxygen to europium for the samples which contain 2, 3, 5, 7 mol% of Rb^+^ ions. Previous studies of similar material agree with our results [40,42]. Moreover, the charge transfer from oxygen to europium seems to be slightly shifted toward the highest wavelength number. However, this could be caused by the incorporation of the rubidium ions into the hydroxyapatite crystal lattice. The peak positions for the excitation and emission spectra are consistent with those expected for Eu^3+^ ions incorporated into calcium(II) ions sites in the hydroxyapatite lattice [40,43,44,45,46].

To evaluate the influence of Rb^+^ ions on the fluorescence lifetime and luminescence kinetics of Eu^3+^ doped hydroxyapatite materials, the luminescence decay profiles were recorded for both series of materials (Figure 7A,B). The luminescence kinetics for both series showed strong correlation between the lifetime of Eu^3+^ ions and the concentration of co-doped Rb^+^ ions in the hydroxyapatite crystal lattice (Figure 7A,B). It should be noted that among samples containing 1 mol% of Eu^3+^ ions, increasing concentrations of rubidium(I) ions led to elongated lifetime of Eu^3+^ ion-related luminescence (Figure 7A). There is a similar correlation for the samples containing 2 mol% of Eu^3+^ ions. However, this tendency is limited to samples containing 3 mol% or less of co-doped Rb+ ions (Figure 7B). Samples which are co-doped with 5 mol% and 7 mol% of Rb^+^ ions showed a decrease in luminescence relative to the other samples. This may indicate that hydroxyapatite nanopowders exhibit quenching when co-doped with 2 mol% of Eu^3+^ ions and more than 3 mol% of Rb^+^ ions.

### 3.3. Evaluation of Biological Properties

#### 3.3.1. Biocompatibility of Obtained Compounds

Both series of compounds exhibit full cytocompatibility toward mouse osteoblasts after 24 h of incubation. The highest tested concentration 100 µg/mL cells showed viability maintained at 80% and more, which is above IC50 (Figure 8). First series of compound showed that the most intensive proliferation rate for the HAp sample containing 1 mol% of Eu^3+^ and 2 mol% of Rb^+^ ions, where 140% of cell viability is observed at the final concentration of compound 10 µg/mL. A small drop-in proliferation rate is observed when sample concentration increases to 50 and 100 µg/mL. However, viability is still maintained well above 100% relative to the negative control. The optimal concentration of nanoparticles for mouse osteoblasts viability is 10 µg/mL for HAp doped with 1 mol% of Eu^3+^ and 2 mol% of Rb^+^ ions, Hap doped with 1 mol% of Eu^3+^ and 3 mol% of Rb^+^ ions, and HAp doped with 1 mol% of Eu^3+^ 7 mol% of Rb^+^ ions (Figure 8A). While concentration 50 µg/mL is suitable for cells for almost all tested powder materials. For the second series of tested materials 10 µg/mL seems to be also the most adequate to promote cells viability. For the second series of doped HAp compounds, concentrations of 10 µg/mL-maintained cell viability at or above 120% (Figure 8B). Our results fully agree with existing studies where hydroxyapatite doped with Rb^+^ ions were obtained and promoted proliferation of MG-63 osteosarcoma cell line in the concentration of 100 µg/mL [47]. However, these compounds and their concentration should be chosen carefully, as one must consider the corresponding increased viability of cancer cells. Another study also confirmed the biocompatibility of rubidium-coated mesoporous glass ceramics and corroborated enhanced adhesion and proliferation of human bone marrow mesenchymal stem cells (HBMSCs) [48]. Similarly, our study confirmed full cytocompatibility toward normal bone cell line, which indicates that our powder materials can be used in future studies, such as and in vivo mouse model.

In addition to the viability assessment of nanopowders, the influence of rubidium ions on proliferation rate of mouse osteoblasts cell lines was investigated (Figure 9). Various concentrations of RbCl were used to evaluate 7F2 cell line viability after 24 h of incubation. It appeared that all tested concentrations of RbCl solution positively affected viability of 7F2 cell line relative to untreated cells, which were established as negative control (Figure 9). Cells treated with 1 mM of RbCl showed around 140% viability, and a gradual increase in viability was observed with increasing RbCl concentration increased. Remarkably, 7F2 cells treated with the highest tested concentration of 5 mM RbCl exhibited viability above 200% relative to the negative control. Furthermore, our results confirmed those of previous studies, which tested RbCl toward primary murine bone marrow monocytes (BMMs) [14]. This research also illustrated the impact of RbCl on osteoclastogenesis and osteoblastogenesis by affecting Jnk and p38-mediated NF-κB activation. By the inhibition of RANKL-induced expression, RbCl attenuate osteoclast marker genes and impairs osteoclastogenesis. Despite the RbCl elevated ALP (alkaline phosphatase) activity and therefore the process of mineralization on both in vitro and in vivo level [14].

Cell morphology and the ratio of live and dead mouse osteoblasts cells were visualized after 24 h incubation with both series of obtained nanopowders having a final concentration of 100 µg/mL (Figure 10). The results of this experiments agreed with the results obtained in the MTT viability assay and clearly showed that morphology and live/dead ratio of 7F2 cell line is the same as those of cells observed in the negative control (untreated cells). Therefore, captured images confirmed results of previously conducted viability assay. Furthermore, unaltered spindle shape of cells and undisturbed cell membrane are comparable to those in the negative control. This indicates full cytocompatibility of synthesized nanopowders (Figure 10).

#### 3.3.2. Evaluation of Hemocompatibility

The hemocompatibility of obtained nanopowders was established in two steps. First, the hemolysis assay was conducted. This was carried out by incubating the purified red blood cells with two different final concentrations of nanopowders: 50 µg/mL and 100 µg/mL. To established positive control, red blood cells were incubated with 10% SDS solution (100% hemolysis). A negative control was established by incubating red blood cells with sterile PBS solution. The incubation period was set up at 24 h at 37 °C. After the established incubation time, the amount of released hemoglobin was measured. As it was confirmed in another study, the 5% of released hemoglobin is commonly acceptable as a naturally occurring hemolysis in the blood system and above this critical point, tested compounds can be classified as potentially harmful toward mammalian erythrocytes [49]. Therefore, hemolysis assay results clearly indicate the hemocompatibility of the presented materials in both tested concentrations (Figure 11). All materials showed hemoglobin release at the level below the critical point of 5% and the results were fully comparable with negative control. Even the standard deviation bars did not cross the critical point (5% of hemolysis) (Figure 11A,B).

To confirm the hemocompatibility of the tested materials, the morphology of red blood cells was visualized after 24 h of incubation by preforming blood smear. The results of this experiment confirmed the results from the hemolysis assay. Since the results of hemolysis assay showed hemocompatibility of both tested concentrations (50 and 100 µg/mL), cell morphology visualization was performed after treatment with the highest tested concentration of obtained nanomaterials. No morphological changes of erythrocytes were observed after incubation with both series of doped HAp samples at the final concentration of 100 µg/mL. the smooth and round shape of the erythrocytes exhibited no membrane disruption or shape alteration. These results are consistent with the negative control (Figure 12). Despite differences between the sedimentation rate of the hydroxyapatite-based materials (in distillated water 12.320 ± 1.003 mm/min—35.130 ± 2.147 mm/min) and red blood cells (in adult sheep erythrocytes, the sedimentation rate was estimated at 0.0333 to 0.0417 mm/min), which could contribute to friction and mechanical damage of erythrocytes, no harmful effect was observed [50,51].

## 4. Conclusions

This paper presents the structural characterization, luminescence, and biological properties of nanopowder hydroxyapatite materials co-doped with 1 mol% and 2 mol% of Eu^3+^ ions and various concentrations of Rb^+^ ions (0.5; 1; 2; 3; 5; 7 mol%). The samples were obtained via the hydrothermal method and thermally treated at 500 °C. They exhibited hydroxyapatite hexagonal structure. However, the structural results (XRPD—X-ray powder diffraction) indicated a slight signal from a secondary phase in samples containing more than 2 mol% of Rb^+^ ions. XRD in conjunction with ICP-OES confirmed that the signal of the secondary phase is RbH_2_PO_4_. Therefore, we confirmed that Eu^3+^ ions do not influence the hydroxyapatite lattice. Elevated molar concentrations of Rb^+^ ions, however, lead to the formation of a secondary phase. On the other hand, FT-IR spectra of obtained materials are consistent with spectra for pure hydroxyapatite powder materials. Hence, hydroxyapatite structure remains but it is contaminated with the secondary phase, identified as RbH_2_PO_4_. The luminescence study showed the characteristic red-orange emission spectra of Eu^3+^ ions incorporated into the materials. While distinctive emission of europium(III) remains consistent among all samples, it can be noted that 0–1, 0–2, and 0–4 transitions increase with higher concentration of Rb^+^ dopant ions. These results reflect those obtained by measuring Eu^3+^ lifetime. Our study confirmed that incorporation of rubidium(I) ions into hydroxyapatite lattice leads to elongation of the luminescence lifetime of Eu^3+^ ions in the matrix. This can be promising factor when considering applications of the obtained compounds in the bioimaging field. Clearly, further investigation is required, especially considering in vitro and in vivo bioimaging tests. Finally, cytotoxicity and hemolysis assays results confirmed the biocompatibility and hemocompatibility of the synthesized samples toward mouse osteoblasts and sheep erythrocytes. This was confirmed for the highest tested concentrations of our nanopowders. Yet, further cytotoxicity tests should be provided in future research. This should be conducted to evaluate the compatibility of synthesized compounds with different cell lines such as dermal fibroblasts or keratinocyte cell lines. More importantly, these materials should be tested against cancer cell lines to exclude potential cancerogenous features. Hydroxyapatite materials co-doped with 1 mol% and 2 mol% of Eu^3+^ ions and various concentration of Rb^+^ ions are promising materials for biomedical applications, especially with respect to bone regenerative medicine.

## Figures and Tables

**Figure 1 nanomaterials-12-04475-f001:**
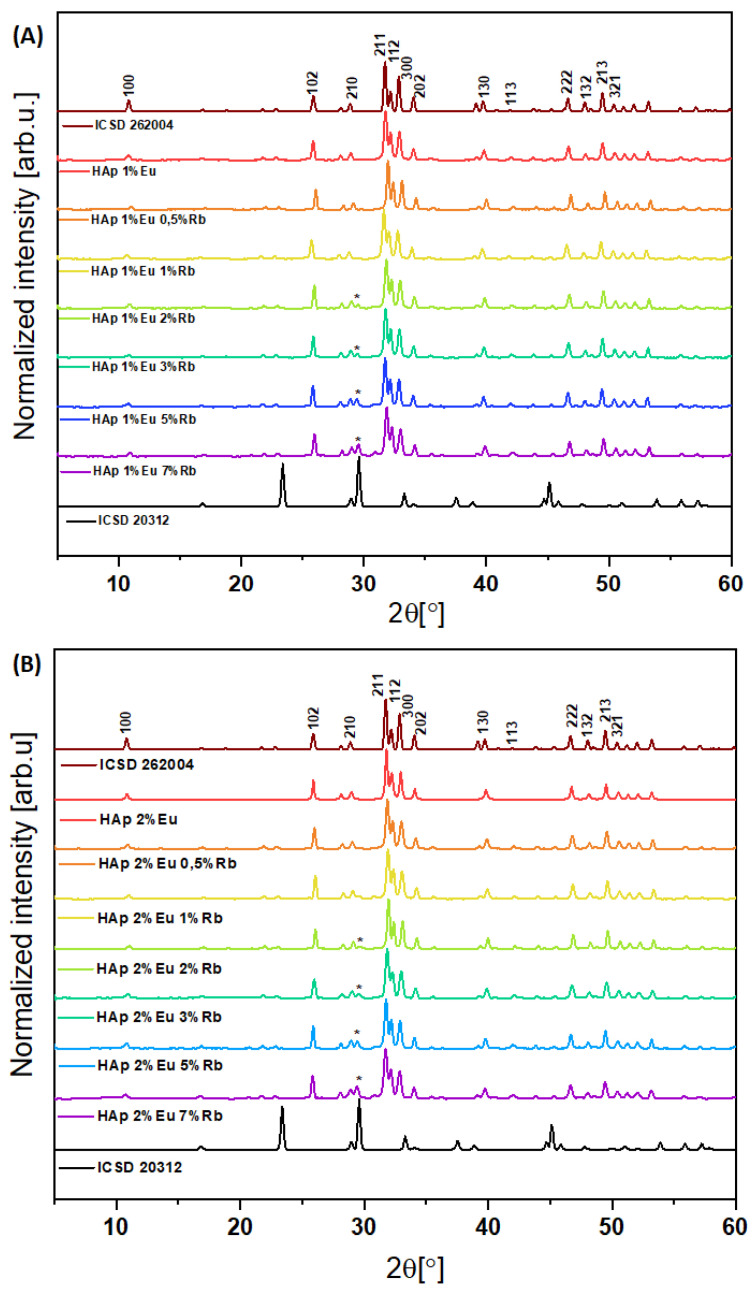
XRD results of Ca_10 − (0.1 + x)_Eu_0.1_Rb_x_(PO_4_)_6_(OH)_2_ (**A**), and Ca_10 − (0.2 + x)_Eu_0.2_Rbx(PO_4_)_6_(OH)_2_ (**B**), where x equals 0.5, 1, 2, 3, 5, 7 mol% of Rb+ ions. The obtained materials were thermally treated at a temperature of 500 °C for 3 h. The XRD results are compared with the ICSD database hydroxyapatite. (*) RbH_2_PO_4_ (ICDS 20312) [18,19].

**Figure 2 nanomaterials-12-04475-f002:**
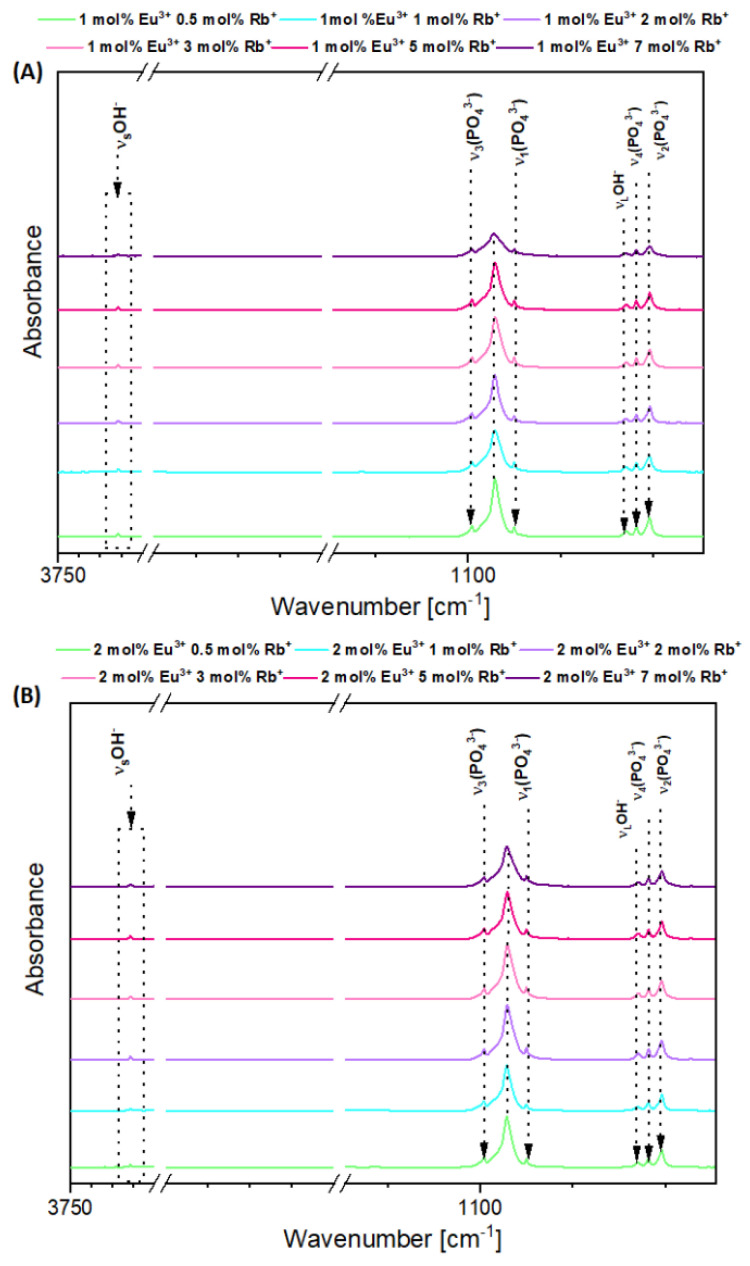
FT-IR spectra of Ca_10 − (0.1 + x)_Eu_0.1_Rb_x_(PO_4_)_6_(OH)_2_ (**A**) and Ca_10 − (0.2 + x)_Eu_0.2_Rb_x_(PO_4_)_6_(OH)_2_ (**B**), where x equals 0.5, 1, 2, 3, 5, 7 mol% of Rb^+^ ions.

**Figure 3 nanomaterials-12-04475-f003:**
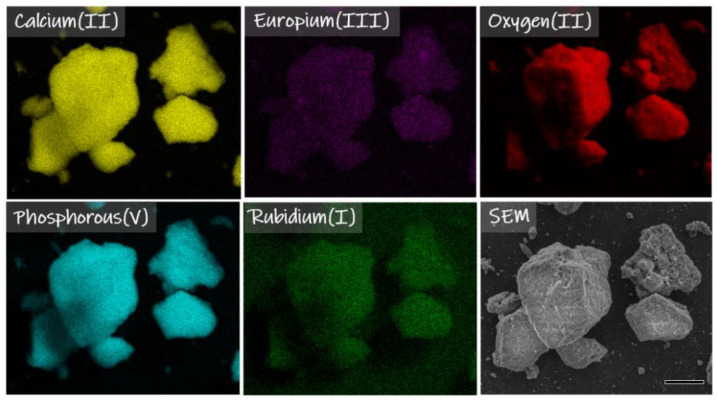
SEM image and SEM-EDS mapping of Ca_9.1_Eu_0.2_Rb_0.7_(PO_4_)_6_(OH)_2_ compound. Images show the elemental composition of the sample which contains calcium, phosphorous, europium, rubidium, and oxygen. Scale bar equals 10 µm.

**Figure 4 nanomaterials-12-04475-f004:**
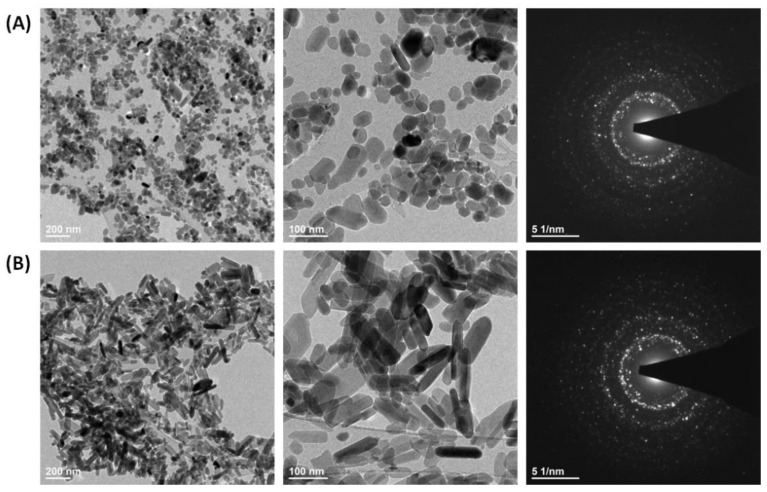
TEM images and SAED images of (**A**) Ca_9.8_Eu_0.1_Rb_0.1_(PO_4_)_6_(OH)_2_ (**B**) Ca_9.7_Eu_0.1_Rb_0.2_(PO_4_)_6_(OH)_2_.

**Figure 5 nanomaterials-12-04475-f005:**
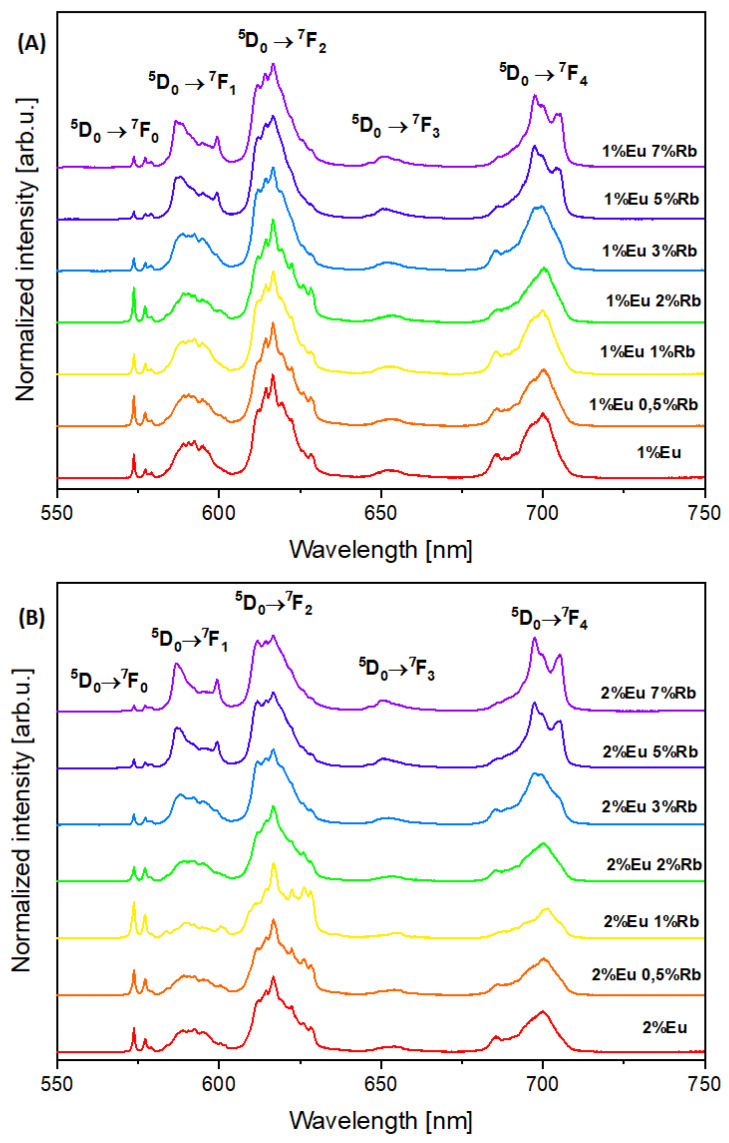
Emission spectra of (**A**) Ca_10 − (0.1 + x)_Eu_0.1_Rb_x_(PO_4_)_6_(OH)_2_ and (**B**) Ca_10 − (0.2 + x)_Eu_0.2_Rb_x_(PO_4_)_6_(OH)_2_, where x equals 0.5, 1, 2, 3, 5, 7 mol% of Rb^+^ ions.

**Figure 6 nanomaterials-12-04475-f006:**
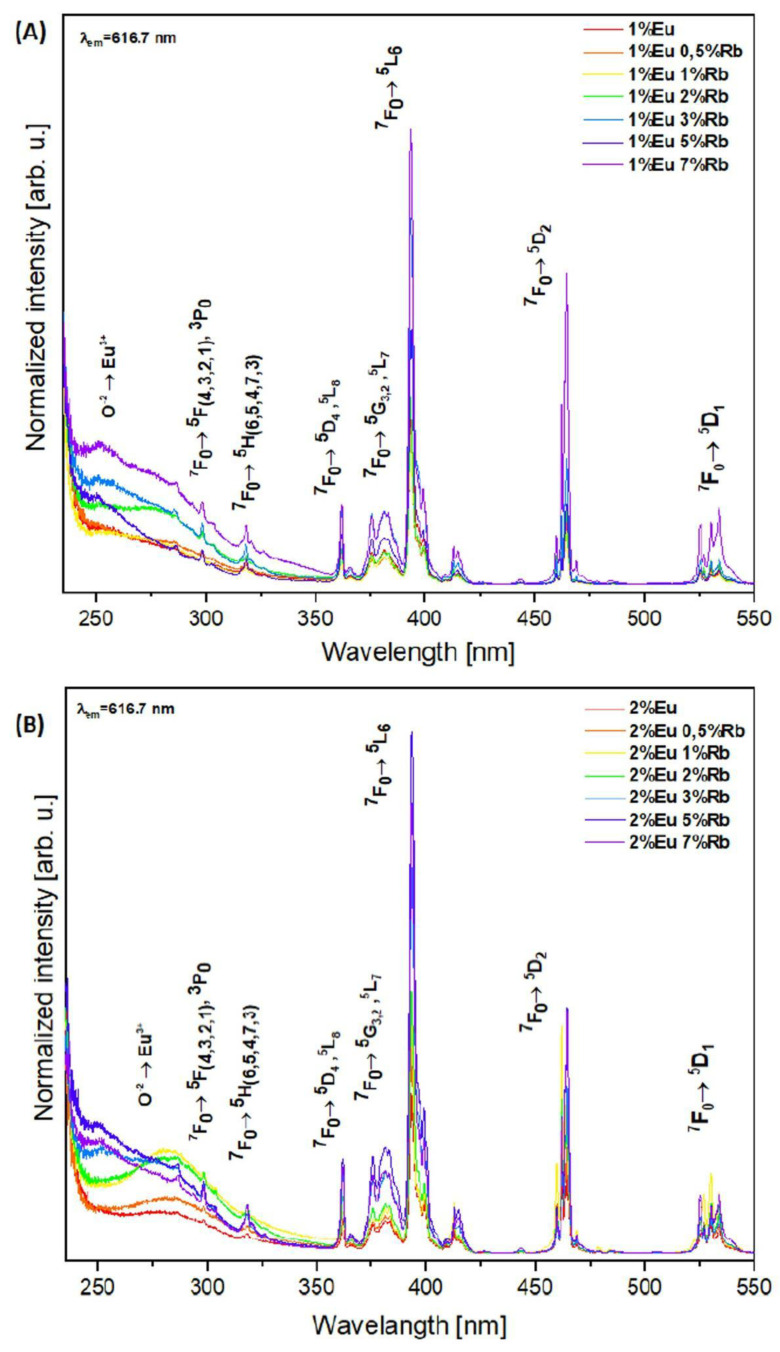
Excitation spectra of (**A**) Ca_10 − (0.1 + x)_Eu_0.1_Rb_x_(PO_4_)_6_(OH)_2_ and (**B**) Ca_10 − (0.2 + x)_Eu_0.2_Rb_x_(PO_4_)_6_(OH)_2_, where x equals 0.5, 1, 2, 3, 5, 7 mol% of Rb^+^ ions.

**Figure 7 nanomaterials-12-04475-f007:**
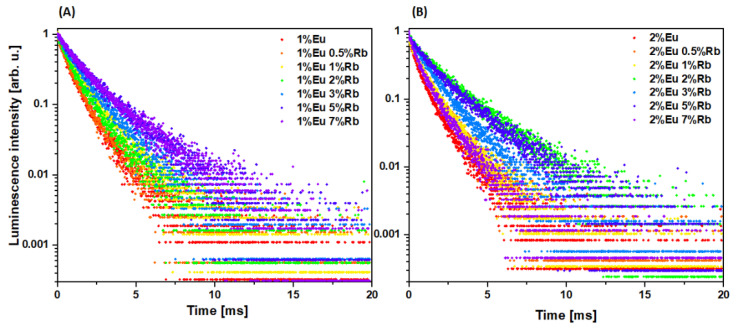
Luminescence decay profiles of (**A**) Ca_10 − (0.1 + x)_Eu_0.1_Rb_x_(PO_4_)_6_(OH)_2_ and (**B**) Ca_10 − (0.2 + x)_Eu_0.2_Rb_x_(PO_4_)_6_(OH)_2_, where x equals 0.5, 1, 2, 3, 5, 7 mol% of Rb^+^ ions.

**Figure 8 nanomaterials-12-04475-f008:**
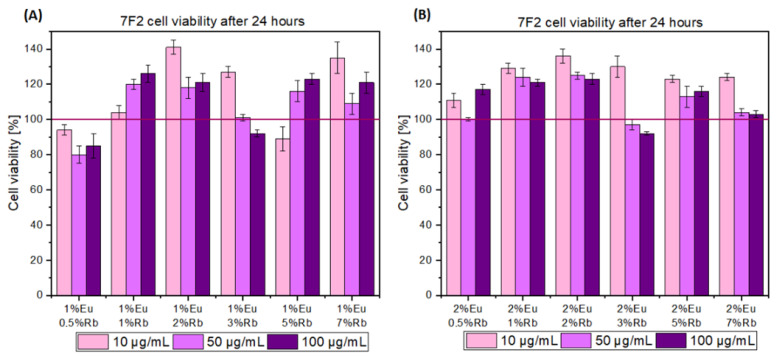
MTT cytotoxicity assay of (**A**) Ca_10 − (0.1 + x)_Eu_0.1_Rb_x_(PO_4_)_6_(OH)_2_ and (**B**) Ca_10 − (0.2 + x)_Eu_0.2_Rb_x_ (PO_4_)_6_(OH)_2_, where x equals 0.5, 1, 2, 3 mol% of Rb^+^ ions. The final concentration of the tested compounds was established at 10 µg/mL, 50 µg/mL, and 100 µg/mL. Red bar marked as 100% of cell viability in the negative control probe.

**Figure 9 nanomaterials-12-04475-f009:**
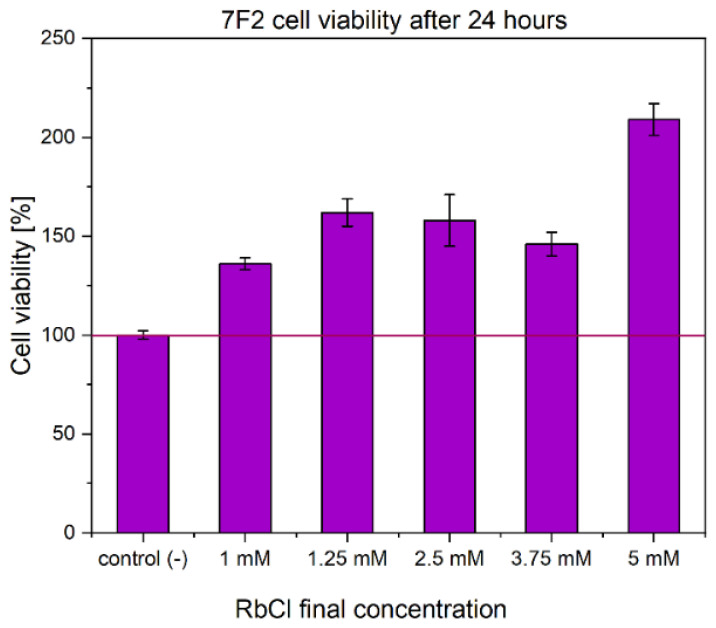
MTT cytotoxicity assay of RbCl towards mouse osteoblasts. Red bar marked as 100% of cell viability in the negative control probe.

**Figure 10 nanomaterials-12-04475-f010:**
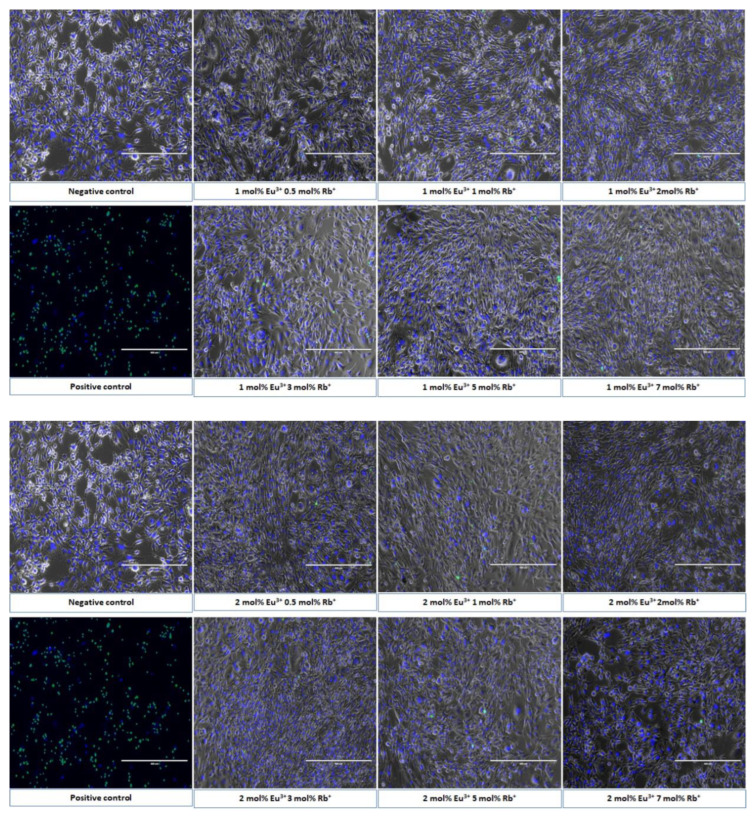
Mouse osteoblasts morphology after 24 h of incubation with both series of compounds Ca_10 − (0.1 + x)_Eu_0.1_Rb_x_(PO_4_)_6_(OH)_2_ and Ca_10 − (0.2 + x)_Eu_0.2_Rb_x_(PO_4_)_6_(OH)_2_, where x = 0.5; 1; 2; 3; 5; 7 mol%) at the final concentration 100 µg/mL. The morphology was compared with negative control (PBS buffer) and positive control (1% H_2_O_2_). Scale bar equals 400 µm.

**Figure 11 nanomaterials-12-04475-f011:**
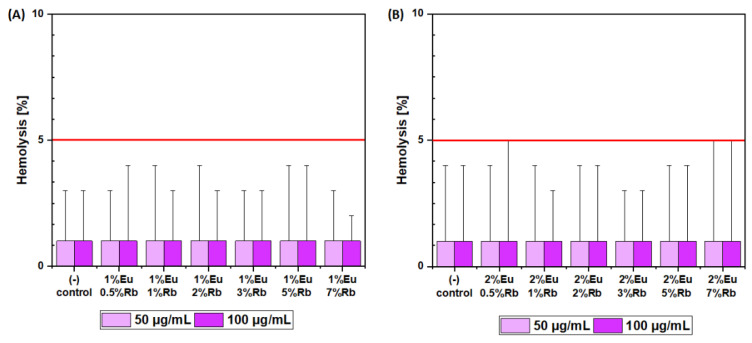
Hemoglobin release after 24 h incubation in (**A**) Ca_10 − (0.1 + x)_Eu_0.1_Rb_x_(PO_4_)_6_(OH)_2_ and (**B**) Ca_10 − (0.2 + x)_Eu_0.2_Rb_x_(PO_4_)_6_(OH)_2_, where x equals 0.5, 1, 2, 3, 5, 7 mol% of Rb^+^ ions. The red line is equal to 5% of physiological hemoglobin release. The results were compared with red blood cells treated with PBS buffer (1% of hemolysis—negative control) and 1% SDS (100% of hemolysis—positive control).

**Figure 12 nanomaterials-12-04475-f012:**
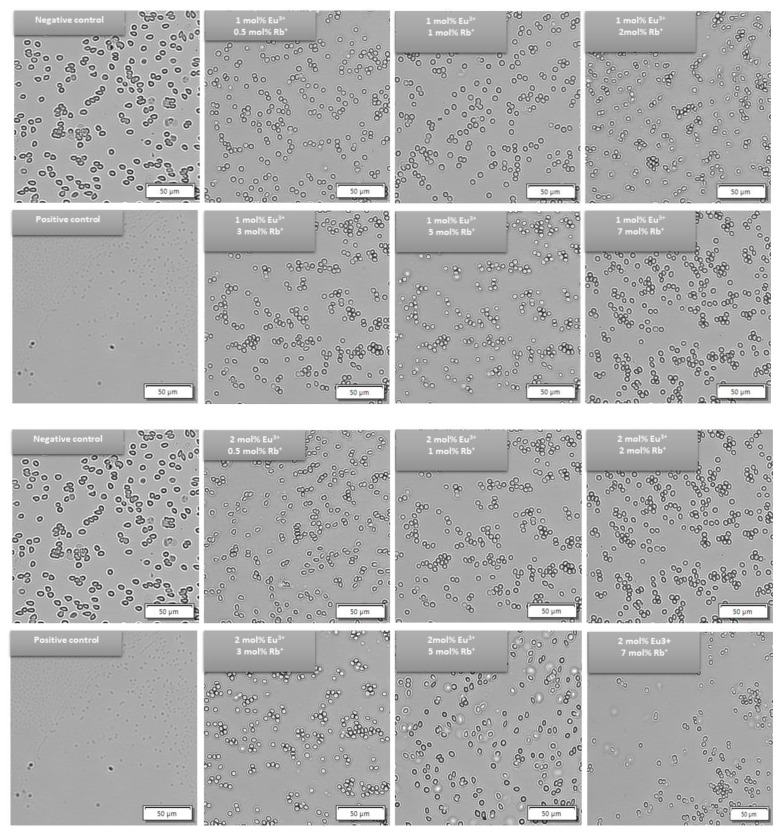
Red blood cell morphology after 24 h of incubation both series of compounds Ca_10 − (0.1 + x)_ Eu_0.1_Rb_x_(PO_4_)_6_(OH)_2_ and Ca_10 − (0.2 + x)_Eu_0.2_Rb_x_(PO_4_)_6_(OH)_2_, where x = 0.5; 1; 2; 3; 5; 7 mol%) at the final concentration 100 µg/mL. The morphology of red blood cells was compared with negative control (PBS buffer) and positive control (1% SDS).

**Table 1 nanomaterials-12-04475-t001:** The elemental contents in the selected nanomaterial samples of Ca_10 − (0.2 + x)_Eu_0.2_Rb_x_(PO_4_)_6_(OH)_2_, where x = 1, 3, 5, 7) based on ICP-OES measurements.

Sample	ICP OES Technique Results			
	n Ca [mol]	n Eu [mol]	n Rb [mol]	n P [mol]
Ca_9.7_Eu_0.2_Rb_0.1_(PO_4_)_6_(OH)_2_	9.66	0.20	0.14	7.01
Ca_9.5_Eu_0.2_Rb_0.3_(PO_4_)_6_(OH)_2_	9.43	0.20	0.37	7.17
Ca_9.3_Eu_0.2_Rb_0.5_(PO_4_)_6_(OH)_2_	9.20	0.20	0.60	7.27
Ca_9.1_Eu_0.2_Rb_0.7_(PO_4_)_6_(OH)_2_	9.00	0.20	0.81	7.44

## Data Availability

Data are available from the authors upon request.

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
