# Peer review of "Structural, Spectroscopic, and Biological Characterization of Novel Rubidium(I) and Europium(III) Co-Doped Nano-Hydroxyapatite Materials and Their Potential Use in Regenerative Medicine"

_nanomaterials, 2022, doi:10.3390/nano12244475_

Round 1

Reviewer 1 Report

The manuscript "Structural, spectroscopic, and biological characterization of  novel rubidium(I) and europium (III) co-doped nano-hydroxyapatite materials and their potential use in regenerative medicine" by Nowak et al. aims to present the synthesis, characterisation and application of new Rb and Eu doped apatitic materials.

A thorough revision is needed before the manuscript could be considered for publication, focusing on the following points:

1. Introduction. Some supplementary info regarding other works should be provided. This would also be useful for the discussion chapter (see further comments)

2. The ICP-OES method does not differentiate between ions of the same elements. Please re-write the methods chapter accordingly

3. Results - describing the XRD patterns as "signals" is incorrect ("signals in the range from 32º to 34º..."). Please use "characteristic peaks", etc. This is the case for the entire presentation of XRD results. Also, when comparing the results or attributing the characteristic peak to a structure, please provide ICDD no. (for the secondary phase). The explanation for the apparition of the secondary phase is relatively simple, as with the increase of the Rb precursor, upon the synthesis is a higher concentration of secondary phase, which translates into a higher relative intensity of the characteristic peak. 

Also, in fig. 1 caption it should be stated to which peak were the values normalized (usually is the highest intensity one, but this should be stated). The identification of each peak by hkl values should be provided (table, eventually as Supplementary Material in order not to overcrowd the figure)

4. Table 1 - The ICP-OES usually presents results as total element (not particular ions) in ppm. Why did the authors chose to present the results as mols?

5. The discussion chapter (especially regarding biological properties) should compare the results with literature data

6. Conclusions section: "secondary phase signal". I suggest to rephrase as "the secondary phase was identified by XRD as...". Also, in this section, please provide a clear take home message of the work   

Minor issues:

- check the manuscript for typos and language errors

- check the manuscript for the correct use of subscript and superscript  

Author Response

Dear Editor,

We would like to express our sincerest gratitude to the reviewers for their thoughtful evaluation of our manuscript. We have considered all raised questions, and carefully revised the manuscript according to the reviewers’ comments.  Here follows the revised manuscript and detailed responses to these comments. Moreover, all changes we have made to the original manuscript are marked with red colour in the text.

Reviewer 1:

The manuscript "Structural, spectroscopic, and biological characterization of  novel rubidium(I) and europium (III) co-doped nano-hydroxyapatite materials and their potential use in regenerative medicine" by Nowak et al. aims to present the synthesis, characterization and application of new Rb and Eu doped apatite materials.

A thorough revision is needed before the manuscript could be considered for publication, focusing on the following points:

Question 1: Introduction. Some supplementary info regarding other works should be provided. This would also be useful for the discussion chapter (see further comments)

Response: It has been provided and corrected.

Question 2:  The ICP-OES method does not differentiate between ions of the same elements. Please re-write the methods chapter accordingly

Response: Thank you for your comment on the ICP-OES methodology. Indeed this method does not differentiate ions of the same elements, therefore this section has been corrected.

Question 3: Results - describing the XRD patterns as "signals" is incorrect ("signals in the range from 32º to 34º..."). Please use "characteristic peaks", etc. This is the case for the entire presentation of XRD results. Also, when comparing the results or attributing the characteristic peak to a structure, please provide ICDD no. (for the secondary phase). The explanation for the apparition of the secondary phase is relatively simple, as with the increase of the Rb precursor, upon the synthesis is a higher concentration of secondary phase, which translates into a higher relative intensity of the characteristic peak. 

Response: The ICSD number of secondary phase has been added in the Figure 1 and additional explanation of the secondary phase occurring has been also provided.

Question 3a: Also, in fig. 1 caption it should be stated to which peak were the values normalized (usually is the highest intensity one, but this should be stated). The identification of each peak by hkl values should be provided (table, eventually as Supplementary Material in order not to overcrowd the figure)

Response: The most distinctive hkl values for hydroxyapatite ICSD 262004 have been added in the Figure 1. Also, the XRD diffractograms were normalized via using OriginPro 2021 software, normalization was set as [0,1] normalization. We decided that normalization according to only one peak will disturbed results perception. Besides, this type of normalization this is too much interference with the results.

Question 4: Table 1 - The ICP-OES usually presents results as total element (not particular ions) in ppm. Why did the authors chose to present the results as mols?

Response: The ICP-OES results are indeed usually presented in ppm. However we decided to calculate ppm content of each element to molar content. It is better to present the ICP-OES results in such form according to the main text of the manuscript the chemical formulas and dopants composition in materials are presented in molar content.

Question 5:  The discussion chapter (especially regarding biological properties) should compare the results with literature data

Response: Thank you for pointing this out, however “Conclusion” contains the summary of our results and, as you can see, in this section only description of our results are presented. We did describe and discuss the results in the section “Results and discussion” well. Supporting the “Conclusion” part with the same references as we did in the section “Result and discussion” is pointless.

Question 6:  Conclusions section: "secondary phase signal". I suggest to rephrase as "the secondary phase was identified by XRD as...". Also, in this section, please provide a clear take home message of the work   

Response: We would like that this particular sentence to remain just as it was written. Besides, it is rather obvious that secondary phase has been identified by X-Ray technique.

Minor issues:

- check the manuscript for typos and language errors

Response: Manuscript typos and language errors have been corrected.

- check the manuscript for the correct use of subscript and superscript  

Response: Manuscript has been checked and all subscript and superscript have been corrected.

Reviewer 2 Report

In my opinion, a very well presented study. Perhaps you should translate into English (if possible) the message displayed on each component of Figure 43.

I am understanding the mentioned text is "an image containing a map automatically generated description", could be introduced in the legend of the figure.

Author Response

Dear Editor,

We would like to express our sincerest gratitude to the reviewers for their thoughtful evaluation of our manuscript. We have considered all raised questions, and carefully revised the manuscript according to the reviewers’ comments.  Here follows the revised manuscript and detailed responses to these comments. Moreover, all changes we have made to the original manuscript are marked with red colour in the text.

Reviewer 2:

Question 1: In my opinion, a very well presented study. Perhaps you should translate into English (if possible) the message displayed on each component of Figure 4.

Response: Thank you for your comment. The Figure 4 has been corrected.

Question 2: I am understanding the mentioned text is "an image containing a map automatically generated description", could be introduced in the legend of the figure.

Response: Additional information related to the SEM-EDS mapping technique has been added in the main text, in section “structural characterization”. Line 120.

Reviewer 3 Report

The manuscript written by Nowak et al.  aimed to obtain and investigate hydroxyapatite (HAp) based compounds co-doped with Eu3+ and Rb+ ions as potential materials for regenerative medicine.  This study is interesting and attractive for the readers in the field of biomaterials research, but some points needs to be clarified before publication.

Specific comments:

1.      The abstract should be rewritten. In the present form is a presentation of the characterization techniques.

2.      The manuscript should be checked for errors (see rows 181, 213) 

Author Response

Dear Editor,

We would like to express our sincerest gratitude to the reviewers for their thoughtful evaluation of our manuscript. We have considered all raised questions, and carefully revised the manuscript according to the reviewers’ comments.  Here follows the revised manuscript and detailed responses to these comments. Moreover, all changes we have made to the original manuscript are marked with red colour in the text.

Reviewer 3:

The manuscript written by Nowak et al.  aimed to obtain and investigate hydroxyapatite (HAp) based compounds co-doped with Eu3+ and Rb+ ions as potential materials for regenerative medicine.  This study is interesting and attractive for the readers in the field of biomaterials research, but some points needs to be clarified before publication.

Specific comments:

  1. The abstract should be rewritten. In the present form is a presentation of the characterization techniques.

 Response: Thank you for your comment. This section has been properly rewritten.

  1. The manuscript should be checked for errors (see rows 181, 213)

Response: All errors have been corrected.

Reviewer 4 Report

The submitted manuscript is nicely written, presents a lot of new results that are properly discussed and the conclusions are correctly drawn. The topic of this study is focused on the physicochemical and biological analysis of the newly obtained materials (co-doped HAp).  However, I would recommend some revisions, details are listed below.

In the title it should be “europium(III)” without the space.

Line 30 – “is the osteoblast which are” – grammatically inconsistent

Introduction is a little bit too long, especially the part in lines 28-45

Lines 55-63, multiple errors here. There should be no space between the () and the element, i.e. “Europium(III)” is a proper way.

Line 91, it should be “(NH4)2HPO4”. A lot of similar errors can be found in that section.

Page 6, PXRD results, the Authors should perform the powder refinement, either Pawley or Rietveld to create and optimize powder diffraction simulation parameters. That way the Authors will obtain the average crystal structures, determine the unit cell parameters and crystallite size. This doesn’t require any additional measurements but only application of software with the already recorded PXRD files. In my opinion, the determination of unit cell parameters and crystallite size will be very beneficial for this work, especially since the Authors have not used any other empirical methods to determine the average crystallite size.

Figure 2, the spectra can be edited even more to cut out the range between the 4000 and 3650 cm-1 as well as 3500 and 3000 cm-1. Also, the part with the hydroxyl groups should have its intensity increased for the better visualization of the shape of the signal.

Table 1, the determined number of P moles is larger than 7 in each case, while, according to the formula, it should be 6. Could you please comment on this?

Also, according to the formulas presented in Table 1, the total charge of the sample is not zero, i.e. Ca9.7Eu0.2Rb0.1(PO4)6(OH)2  9.7*2+0.2*3 + 0.1*2 -6*3 – 2*1 = 0.2. Please comment on this.

Figure 11, this figure can be edited too. The y-axis should be limited to 5% as the area above this level is not used.

Author Response

Dear Editor,

We would like to express our sincerest gratitude to the reviewers for their thoughtful evaluation of our manuscript. We have considered all raised questions, and carefully revised the manuscript according to the reviewers’ comments.  Here follows the revised manuscript and detailed responses to these comments. Moreover, all changes we have made to the original manuscript are marked with red colour in the text.

Reviewer 4:

The submitted manuscript is nicely written, presents a lot of new results that are properly discussed and the conclusions are correctly drawn. The topic of this study is focused on the physicochemical and biological analysis of the newly obtained materials (co-doped HAp).  However, I would recommend some revisions, details are listed below.

Question 1: In the title it should be “europium(III)” without the space.

Response: Thank you for this comment. The title has been corrected.

Question 2: Line 30 – “is the osteoblast which are” – grammatically inconsistent

Response: This part has been removed from introduction section.

Question 3: Introduction is a little bit too long, especially the part in lines 28-45

Response: The first part of the introduction, where function of extracellular matrix and its components, has been removed.

Question 4: Lines 55-63, multiple errors here. There should be no space between the () and the element, i.e. “Europium(III)” is a proper way.

Response: Thank you for noticing minor errors and misspelling in the text. We corrected each writing mistakes.

Question 5: Line 91, it should be “(NH4)2HPO4”. A lot of similar errors can be found in that section.

Response: All chemical formulas have been corrected.

Question 6: Page 6, PXRD results, the Authors should perform the powder refinement, either Pawley or Rietveld to create and optimize powder diffraction simulation parameters. That way the Authors will obtain the average crystal structures, determine the unit cell parameters and crystallite size. This doesn’t require any additional measurements but only application of software with the already recorded PXRD files. In my opinion, the determination of unit cell parameters and crystallite size will be very beneficial for this work, especially since the Authors have not used any other empirical methods to determine the average crystallite size.

Response: We agree with the Reviewer regarding the powder refinement and normally, we would have done it.

Anyway, in the case of the obtained materials, it is difficult to compare the results of the refinement in the series because not all of the obtained materials are phase pure. In order to solve a structure from powder data it is necessary to use some physical methods, therefore, there have been used the XRDP, TEM and SEM as well as ICP-OES methods to resolve structure and morphology of the obtained materials. The average crystallites' size could be directly determined from TEM images.

Question 7: Figure 2, the spectra can be edited even more to cut out the range between the 4000 and 3650 cm-1 as well as 3500 and 3000 cm-1. Also, the part with the hydroxyl groups should have its intensity increased for the better visualization of the shape of the signal. -

Response: The Figure 2 has been corrected.

Question 8: Table 1, the determined number of P moles is larger than 7 in each case, while, according to the formula, it should be 6. Could you please comment on this?

Response: Thank you for point this out. This issue is explained in line 301-304

Question 9: Also, according to the formulas presented in Table 1, the total charge of the sample is not zero, i.e. Ca9.7Eu0.2Rb0.1(PO4)6(OH)2  9.7*2+0.2*3 + 0.1*2 -6*3 – 2*1 = 0.2. Please comment on this.

Response: The resulting charge imbalance is quite typical for compounds of this type doped with cations of a different charge than the base element (here - Ca2+ ion). The cationic vacancies are formed by replacing Ca2+ ions with Eu3+ ions with the higher charge and can result in a non-zero net charge. This issue has been addressed in the following articles: [1] [2] [3].

[1] Cazalbou, S. et al. “Adaptative physico-chemistry of bio-related calcium phosphates.” Journal of Materials Chemistry 14 (2004): 2148-2153. doi: 10.1039/B401318B.

[2] Tite T, Popa A-C, Balescu LM, Bogdan IM, Pasuk I, Ferreira JMF, Stan GE. Cationic Substitutions in Hydroxyapatite: Current Status of the Derived Biofunctional Effects and Their In Vitro Interrogation Methods. Materials. 2018; 11(11):2081. https://doi.org/10.3390/ma11112081.

[3] Targonska S, Wiglusz R.J. Investigation of Physicochemical Properties of the Structurally Modified Nanosized Silicate-Substituted Hydroxyapatite Co-Doped with Eu3+ and Sr2+ Ions. Nanomaterials (Basel). 2020 Dec 24;11(1):27. doi: 10.3390/nano11010027.

Question 10: Figure 11, this figure can be edited too. The y-axis should be limited to 5% as the area above this level is not used.

Response: Thank you for your comment on the graph. Figure 11 has been corrected.

Round 2

Reviewer 1 Report

The authors addressed the points raised by the reviewer. The manuscript can be accepted in the present form.

Reviewer 4 Report

The Authors have significantly improved their manuscript. This version can be accepted for publication 'as it is now'.